# Decision Boundaries and Convex Hulls in the Feature Space that Deep Learning Functions Learn from Images

## Abstract

The success of deep neural networks in image classification and learning can be partly attributed to the features they extract from images. It is often speculated about the properties of a low-dimensional manifold that models extract and learn from images. However, there is not sufficient understanding about this low-dimensional space based on theory or empirical evidence. For image classification models, their last hidden layer is the one where images of each class is separated from other classes and it also has the least number of features. Here, we develop methods and formulations to study that feature space for any model. We study the partitioning of the domain in feature space, identify regions guaranteed to have certain classifications, and investigate its implications for the pixel space. We observe that geometric arrangements of decision boundaries in feature space is significantly different compared to pixel space, providing insights about adversarial vulnerabilities, image morphing, extrapolation, ambiguity in classification, and the mathematical understanding of image classification models.

## 1 Introduction

The process in which deep networks learn to classify images is not adequately understood. In the context of classification, successful learning can be described as learning the similarities and differences between samples of each class. But for images, similarities and differences usually cannot be identified or explained in terms of individual pixels. So, how do models and humans identify similarities and see differences in images? The spatial relationship between groups of pixels and the patterns that are depicted via such pixel groups are instrumental in classifying them by humans and models. If we ask a person why they classify a particular image as a cat, they might point out the specific patterns such as the shape of ears and eyes of the cat. If we ask a radiologist why they classify a tumor as cancerous, they might point out the shape of the tumor and the patterns visible in that region. Analyzing these patterns can be considered feature extraction, and those features, as opposed to individual pixels, would be the ones helpful for learning and classification.

In deep learning, feature extraction is performed via specialized computational tools, i.e., convolutional layers, and it is not easy to disentangle the feature extraction from the learning process as a whole. Often, when a model has good generalization accuracy, one considers that the model has learned some useful features (Chen et al., 2021), but it is not clear what those features are (Berner et al., 2021). This lack of understanding is evident when we consider vulnerability of models to adversarial examples (Shafahi et al., 2019). Sometimes adversarial examples are themselves considered features (Ilyas et al., 2019). Another issue arises when one gives out-of-distribution images to a model, e.g., a model trained for object recognition may classify a radiology image of liver as Airplane with 100% confidence, defying the notion of learning. Despite these shortcomings, deep networks are impressively successful in a wide range of tasks related to image classification, e.g., facial recognition, object recognition, medical imaging. There have been several studies to improve our understanding of what models learn from images, e.g., Xiao et al. (2020) studied the effect of image backgrounds. Several other studies focused on verifying whether models have learned generalizable features (Yadav & Bottou, 2019; Recht et al., 2018; 2019). Neyshabur (2020) used feedforward networks to learn the convolutional filters from scratch. Alain & Bengio (2016) studied linear separability of the classes in intermediate layers of trained networks. Balestriero & Baraniuk (2021) showed that deep neural networks

are spline operators that partition their domain. Recanatesi et al. (2021) studied feedforward networks and concluded that models learn a low-dimensional latent representation from images. This idea is pursued before under a field known as representation learning (Bengio et al., 2013; Oord et al., 2018). There are studies on geometry of data and the separability of classes, e.g., (Mallat, 2016; Cohen et al., 2020; Fawzi et al., 2018; Bronstein et al., 2017). Moreover, specific deep learning architectures are introduced that process images with wavelet scattering (Bruna & Mallat, 2013; Zarka et al., 2020) to provide a way to understand properties of the features learned by the models, e.g., Zarka et al. (2021) studied the Fisher discriminant ratio of learned features.

In this paper, we develop methods to complement the previous work and provide a better understanding of the feature space that deep networks learn from images. We consider the last hidden layer of image classification models as the feature space with least number of features where images of each class are separated from other classes. Our contributions can be summarized as:

1. We develop methods and formulations that can be used to systematically investigate the feature space learned by any trained model. We investigate how images map to the feature space, and how that feature space relates to the pixel space. Finding images in the pixel space that would directly map to particular points and regions in the feature space is an inverse problem involving the trained models, the type of problem that is generally considered hard to solve (Elsayed et al., 2018). We use the homotopy algorithm by Yousefzadeh & O'Leary (2020) to solve our formulations.

2. We study the functional task of models in that feature space and see that testing samples are all outside the convex hull of training set even in a 64-dimensional feature space learned by the models, i.e., functional task of models involve moderate extrapolation. Yousefzadeh (2020) reported that image classification requires extrapolation both in pixel space and in the feature space. More recently, Balestriero et al. (2021) concluded that in high-dimensional space (larger than 100 dimensions), learning always amounts to extrapolation. Our results in this paper show that even in a 64-dimensional space learned by the models, image classification still requires extrapolation.

3. Our method identifies points in the pixel space that would map to decision boundaries and convex hulls in the feature space providing novel insights about the functional performance of models in that space, and the extent of extrapolation. We observe that arrangements of decision boundaries and convex hulls in feature space differ from the pixel space in meaningful ways, not reported in the literature. Our methods can also be used for image morphing.

4. We propose a new method to identify ambiguous and adversarial images based on their relative distance to decision boundaries and the convex hull of training set in the feature space. In the feature space, unlike the pixel space, most testing images are relatively close to the convex hull of training set while far from the decision boundaries. Ambiguous images, however, are close to decision boundaries and far from the convex hull. Adversarial inputs are also recognizably close the decision boundaries of feature space. Moreover, adversarial methods such as DeepFool (Moosavi-Dezfooli et al., 2016) and PGD attacks (Madry et al., 2018) move images towards the convex hull of training set in the feature space.

## 2 Feature space learned by trained models

We consider the feature space in the last hidden layer of trained models. This feature space is the key to successful classification of images and it usually has the least dimensionality compared to other hidden layers. Our trained model is a function denoted by $\mathcal{N}(.)$ that operates on input images and produces an output vector

$$z = \mathcal{N}(x), \tag{1}$$

where each element of $z$ corresponds to one class, and the class(es) with the largest value will be the classification of the model[1]

$$\mathcal{C}(z) = \{i : z_i = \max_k z_k\}. \tag{2}$$

---

[1]For brevity, we may sometimes use $\mathcal{C}(x)$ to denote the classification of the model for $x$, implying that a $z$ has been computed for $x$ and $\mathcal{C}(.)$ has been applied to that $z$.

Domain of $\mathcal{N}$ is denoted by $\Omega$ which would be the pixel space for image classification models. Any given model is trained to recognize a certain number of classes. In our notation, pixel space has $p$ dimensions/pixels and $z$ has $n$ elements/classes.

We use $\Phi$ to denote the feature space in the last hidden layer of $\mathcal{N}$. An input image, $x$, has a mapping to that feature space denoted by $x_\phi$. We can formalize this mapping via our trained model

$$x_\phi = \mathcal{N}_\phi(x), \tag{3}$$

where $\Phi$ has $f$ dimensions. $\mathcal{N}_\phi(.)$ is similar to $\mathcal{N}(.)$ except that it returns the output of the last hidden layer of the model. Similar to pixel space, feature space will also have a domain, $\Omega_\phi$ which would be the range of $\mathcal{N}_\phi(.)$.

After the last hidden layer, the model has a fully connected layer and a softmax layer. Hence, the output of the model, $z$, can be written in terms of the feature space:

$$z = \text{softmax}(x_\phi W_\phi + b_\phi), \tag{4}$$

where $W_\phi$ is the weight matrix for the last fully connected layer, with $f$ rows and $n$ columns, and $b_\phi$ is the bias vector for that layer with $n$ elements. It is sensible to assume $n < f$, i.e., feature space has more dimensions than the number of output classes.

Our following formulations are applicable to any model with any number of features in its hidden layers, i.e., $\mathcal{N}$ can be any trained model. Moreover, one can study the feature space in any of the hidden layers, though, in this work, our focus is on the last hidden layer. To make this more tangible, consider $\mathcal{N}$ to be a standard CNN, pre-trained on CIFAR-10 dataset. Model has a standard residual network architecture (He et al., 2016) with total depth of 20 layers while the last hidden layer has 64 features.[2] It follows that $\Phi$ for this particular model has 64 dimensions. We choose this model because its last hidden layer has fewer features than the standard ResNet-18.

For a given $x$, one can easily compute its corresponding $x_\phi$ (i.e., map $x$ to $\Phi$) by feeding $x$ to the trained model and computing the output of the model's last hidden layer. However, given an arbitrary $x_\phi$, it is not as easy to find its corresponding $x$ in the pixel space. That is, a trained model $\mathcal{N}$, and by extension $\mathcal{N}_\phi$, are not invertible, i.e., there is not an inverse function $\mathcal{N}_\phi^{-1}$ readily available to map an arbitrary $x_\phi$ to the pixel space. Moreover, the mapping from the pixel space to $\Phi$ is not one-to-one.[3]

In Sections 4-5, we will formulate and solve optimization problems to find images (in the pixel space) that would directly map to particular points and regions in the feature space. Before that, let us formulate the decision boundaries of the model in the feature space.

## 3 Decision boundaries in the feature space

An image classification model is a classification function that partitions its domain and assigns a class to each partition (Strang, 2019). Partitions are defined by decision boundaries and so is the model. We can study the decision boundaries and partitions of the model, not just in the pixel domain, but also in the feature space $\Phi$. A point on the decision boundary between classes $i$ and $j$ would be a point that satisfies

$$z_i = z_j, \tag{5}$$

$$z_i \geq z_k, \forall k \notin \{i, j\}. \tag{6}$$

Any point that satisfies the conditions above will be a flip between classes $i$ and $j$, so we call it a flip point (Yousefzadeh & O'Leary, 2020; 2021). We denote flip points by $x^{f(i,j)}$ when they are in the pixel space, and denote them by $x_\phi^{f(i,j)}$ when they are in the feature space.

---

[2]Pre-trained model is available at `https://www.mathworks.com/help/deeplearning/ug/train-residual-network-for-image-classification.html`.

[3]This can be easily verified via any of the pooling layers.

For the purpose of identifying points on the decision boundaries of the model, we can ignore the softmax operation in equation 4 because it only normalizes the values of $z$ to be between 0 and 1, and does not change their order. Therefore, in the following, we will drop the softmax from equation 4 because it does not have an effect on satisfying constraints 5-6. As a result $x_\phi^{f(i,j)}$ should satisfy

$$x_\phi^{f(i,j)} W_\phi(:,i) + b_\phi(i) = x_\phi^{f(i,j)} W_\phi(:,j) + b_\phi(j), \tag{7}$$

$$x_\phi^{f(i,j)} W_\phi(:,i) + b_\phi(i) \geq x_\phi^{f(i,j)} W_\phi(:,k) + b_\phi(k), \ \forall k \notin \{i,j\}, \tag{8}$$

where $W_\phi(:,i)$ denotes the $i^{th}$ column of $W_\phi$.

For a given model, there usually are infinite number of $x_\phi^{f(i,j)}$ satisfying the constraints 7-8, but we may be interested to find the $x_\phi^{f(i,j)}$ that is closest to a particular $x_\phi$. Consider that element $i$ of $z$ has the largest value for the input $x$, i.e., classification of $x$ and $x_\phi$ are $i$. The *closest* flip point to $x_\phi$ between classes $i$ and $j$ is denoted by $x_\phi^{f(i,j),c}$ and obtained via the objective function

$$\min_{x_\phi^{f(i,j),c}} \|x_\phi - x_\phi^{f(i,j),c}\|_2^2. \tag{9}$$

Our feature space $\Phi$ is usually lower bounded by zero because it is the result of convolutional, ReLU, and max pooling layers. Hence, we require

$$0 \leq x_\phi^{f(i,j),c}. \tag{10}$$

The optimization problem defined by objective function 9 subject to constraints 7,8, and 10 is convex, and there are reliable algorithms to solve it. In most cases, it may be strictly convex, making the optimal solution unique. Either way, the minimum distance to decision boundaries (a.k.a. margin) will be a unique value. The minimum distance of $x_\phi$ to the decision boundary between classes $i$ and $j$ is

$$d_\phi^{f(i,j)}(x_\phi) = \|x_\phi - x_\phi^{f(i,j),c}\|_2. \tag{11}$$

For a model with $n$ output classes and for a specific input $x$, mapped to $x_\phi$ and classified as $i$, we can compute its margin in $\Phi$ to all other $n-1$ classes and find out decision boundary of which class is closest to it. We denote the closest margin by

$$d_\phi^{f,min}(x_\phi) = \min_{j \in \{1:n \setminus i\}} d_\phi^{f(i,j)}(x_\phi). \tag{12}$$

Consider, for example, the 2D domain depicted in Figure 1 which has 5 partitions representing 5 different classes. Input $x$ is located in the partition associated with Class 1. This particular input has a margin to each of the other four classes and the minimum margin is to Class 4. Since our optimization problems in $\Phi$ are convex, we can calculate $d_\phi^{f,min}$ precisely and be sure that it actually is the distance to the closest decision boundary.

Let us now consider the ball centered at $x_\phi$ with radius $d_\phi^{f,min}$, and denote it by $\mathcal{B}(x_\phi)$. Such ball may be entirely inside the domain of feature space, $\Omega_\phi$, or it may extend outside the domain, if $x_\phi$ is close to the boundaries of the $\Omega_\phi$ in some dimensions. Either way, classification of $\mathcal{N}$ for the entire region inside the intersection of $\mathcal{B}(x_\phi)$ and $\Omega_\phi$ is guaranteed to be the same as the classification for $x$ and $x_\phi$

$$\forall y_\phi \in (\mathcal{B}(x_\phi) \cap \Omega_\phi) : \mathcal{C}(y_\phi) = \mathcal{C}(x_\phi), \tag{13}$$

i.e., any point in $\Omega_\phi$ that its distance to $x_\phi$ is less than $d_\phi^{f,min}(x_\phi)$ has the same classification as $x_\phi$. For this guarantee, we note that $\Phi$ is a continuous space and the output of $\mathcal{N}$ is Lipschitz continuous with respect to points in $\Phi$. In fact, Lipschitz constant of the model with respect to $\Phi$ would be $\sigma_{max}(W_\phi)$, i.e., the largest singular value of $W_\phi$, since one can prove

$$\|z(x_\phi) - z(y_\phi)\|_2 \leq \sigma_{max}(W_\phi)\|x_\phi - y_\phi\|_2, \tag{14}$$

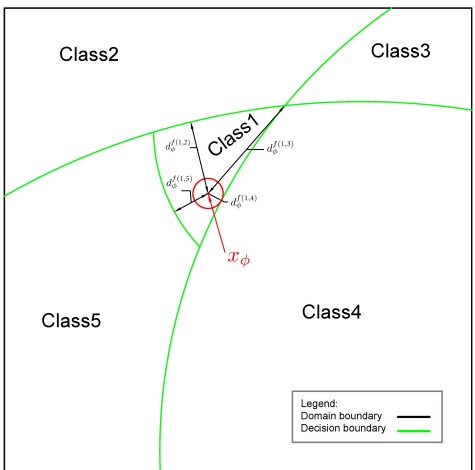

Figure 1: An example 2D domain with 5 partitions. Input $x_\phi$, in the feature space, is located in the partition for Class 1. Its margin to each of the other classes is marked. The red ball is the largest ball centered around $x_\phi$ where every point inside it is guaranteed to have the same classification as $x_\phi$.

for any $x_\phi$ and $y_\phi$ in feature space.

The radius of the ball $\mathcal{B}(x_\phi)$ gives a measure of robustness for the classification of the model with respect to perturbations in feature space. By studying the decision boundaries, one can also design and analyze adversarial inputs in the feature space and then trace them back to the pixel space as we will explore in numerical experiments. In the following two sections, we provide formulations to reveal the relationship between feature space and pixel space.

## 4 Seeking points in the pixel space that would map to particular regions in the feature space

As the first goal, let us find images in the pixel space that would map to particular regions in the feature space. Specifically, we seek to find images in the pixel space that would map to $\mathcal{B}(x_\phi) \cap \Omega_\phi$ around any particular image $x_\phi$ satisfying equation 13. The following constraints will ensure such mapping for $x^{\Omega \to \mathcal{B}(x_\phi)}$

$$\|\mathcal{N}_\phi(x^{\Omega \to \mathcal{B}(x_\phi)}) - x_\phi\|_2 < \lambda \quad , \quad x \in \Omega, \tag{15}$$

where $\lambda$ is the radius of the $\mathcal{B}$ or region of interest.

Many different images (in pixel space) may satisfy the constraint above for a particular $x_\phi$ as we shall see in experimental results. To gain an understanding of the variety of such images, we seek to find the ones that are closest to a reference point, $x^r$, in pixel space. A reference point may be any training or testing image, or any other image such as a completely black or white image. Minimizing the distance to reference point is our objective function

$$\min_{x^{\Omega \to \mathcal{B}(x_\phi)}} \|x^{\Omega \to \mathcal{B}(x_\phi)} - x^r\|_2^2, \tag{16}$$

and our constraint is equation 15. We can solve this optimization problem for various reference points, $x^r$, to gain an understanding of the ball surrounding the sample $x_\phi$.

Unlike our set of optimization problems in Section 3, optimization problems in Sections 4 and 5 may be non-convex because they involve a typically non-convex function $\mathcal{N}_\phi$. Hence, it is important that global optimization algorithms be utilized for solving them. Moreover, issue of vanishing and exploding gradients (Bengio et al., 1994) may arise which is addressed in our previous work.

# 5 Seeking points in the pixel space that would map to particular points in feature space

We now seek points in the pixel space that $\mathcal{N}_\phi$ will directly map them to a particular $x_\phi$. For an input $x^{\Omega \to x_\phi}$, this condition can be formalized as:

$$\mathcal{N}_\phi(x^{\Omega \to x_\phi}) = x_\phi \quad , \quad x^{\Omega \to x_\phi} \in \Omega. \tag{17}$$

The particular $x_\phi$ may be any point of interest in the feature space, for example, a point on a decision boundary, or a point on the boundary of a convex hull.

It is possible that $x^{\Omega \to x_\phi}$ defined by equations 17 is not unique, rather, a region, $\mathcal{S}^{\Omega \to x_\phi}$, in the pixel space (contiguous or not), will all map to a particular point in the feature space. We seek to find the $x_\Omega^{h,\phi}$ that is *closest* to a reference point $x^r$ using the objective function

$$\min_{x^{\Omega \to x_\phi}} \|x^{\Omega \to x_\phi} - x^r\|, \tag{18}$$

subject to constraint 17.

It is sensible to use a reference point that has the same classification as $x_\phi$. In such case, we can impose an additional constraint to ensure $x^{\Omega \to x_\phi}$ and $x^r$ belong to the same partition in pixel space.

$$\exists \pi, \pi : (x^{\Omega \to x_\phi}, x^r) \mid \forall x \in \pi, \mathcal{C}(\mathcal{N}(x)) = \mathcal{C}(\mathcal{N}(x^r)), \tag{19}$$

To verify the additional constraint 19, one needs to verify Lipschitz continuity of $\mathcal{N}$ not just in the feature but also in the pixel space $\Omega$. There are methods to estimate the Lipschitz constant for neural networks (Scaman & Virmaux, 2018). In our empirical experiments, we see that when $x^r$ has the same classification as $x_\phi$, this constraint is automatically satisfied via a direct path.

# 6 Convex hull of training set in feature space

We now turn our attention to geometric properties of training and testing set in the feature space. Mainly, we investigate the geometry of testing samples with respect to the convex hull of training set. Using equation 3, we can map all training samples to $\Phi$ and form their convex hull. $\mathcal{H}_\phi^{tr}$ denotes the convex hull of training set in $\Phi$ while $\mathcal{H}^{tr}$ denotes the convex hull of training set in the pixel space. Furthermore, projection of $x$ to $\mathcal{H}^{tr}$ is denoted by $x^h$, and projection of $x_\phi$ to $\mathcal{H}_\phi^{tr}$ is denoted by $x_\phi^h$.

It is reported that for standard image classification datasets, testing samples are entirely outside $\mathcal{H}^{tr}$ and $\mathcal{H}_\phi^{tr}$. As a result, a model has to extrapolate in order to classify testing samples (Yousefzadeh, 2020; Balestriero et al., 2021). Here, we study the extent of such extrapolation in the feature space and investigate its implications for the pixel space. Particularly, for a given $x$ and its corresponding $x_\phi^h$, we would like to find the least changes in $x$ that would directly map it to $x_\phi^h$. Moreover, using the formulations in previous sections, we will investigate the decision boundaries of the model in feature space with respect to the $\mathcal{H}_\phi^{tr}$, as presented in numerical experiments. Before that, we briefly review the computations necessary to project a point to a convex hull.

## 6.1 Projecting a query point to a convex hull

In the feature space, as in the pixel space, projecting a query point to a convex hull can be performed by solving a convex optimization problem. Here, we briefly review the formulation. While there are off-the-shelf algorithms to solve this problem, Yousefzadeh (2021) has provided a sketching algorithm that may be beneficial for solving the problem faster.

Given a point in the feature space, $x_\phi$, we would like to find the closest point to it on the $\mathcal{H}_\phi^{tr}$. Distance can be measured using any desired norm. Here, we use the 2-norm distance and minimize it via the objective function

$$\min_{x_\phi^h} \|x_\phi^h - x_\phi\|_2^2 \tag{20}$$

Our first constraint relates the solution to the samples in training set

$$x_\phi^h = \alpha \mathcal{D}_\phi, \tag{21}$$

where $\mathcal{D}_\phi$ is the training set, in the feature space, formed as a matrix where rows represent $n$ samples and columns represent $d_\phi$ features. The other two constraints ensure that $x_\phi^h$ belongs to the convex hull of $\mathcal{D}_\phi$.

$$\alpha \mathbb{1}_{n,1} = 1, \tag{22}$$

$$0 \leq \alpha. \tag{23}$$

Minimizing the objective function 20 subject to constraints 21-23 will lead to the point on $\mathcal{H}_\phi^{tr}$ closest to $x_\phi$. Since our optimization problem is convex, there is guarantee to find its solution. We denote this projection with

$$x_\phi^h = \mathcal{P}^h(x_\phi, \mathcal{H}_\phi^{tr}), \tag{24}$$

while distance to $\mathcal{H}_\phi^{tr}$ is denoted by

$$d_\phi^h(x_\phi) = \|x_\phi - x_\phi^h\|_2. \tag{25}$$

Using the optimization problem formulated in Section 5, we may map $x_\phi^h$ back to the pixel space.

# 7 Numerical experiments

We first investigate a single image of CIFAR-10 dataset (Krizhevsky, 2009) in detail and from different perspectives. Later in Section 7.2, we report the larger trends in this dataset.

## 7.1 Insights about one image

Let us consider $x$ to be the first testing sample of dataset shown in Figure 2a. Our model is a standard pre-trained model described in Section 3 and available at the link in footnote 2. We map this image to the feature space of the model to obtain $x_\phi$. Since $x_\phi$ has 64 elements, we can plot it as an 8 by 8 image:

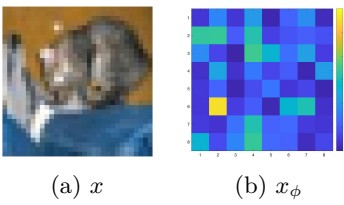

(a) $x$      (b) $x_\phi$

Figure 2: First testing sample in CIFAR-10 dataset **(a)** in pixel space. **(b)** mapping of $x$ to the feature space in the last hidden layer of our trained model.

**Decision boundaries in the feature space.** We first investigate the decision boundaries of the model in the vicinity of $x_\phi$. Classification of the model for this image is Cat. Table 1 shows the margin of $x_\phi$ to each of the other 9 classes.

Table 1: Distance to decision boundaries of each class in the feature space $\Phi$ (sample in Figure 2)

| Class | airplane | car | bird | cat | deer | dog | frog | horse | ship | truck |
|-------|----------|-----|------|-----|------|-----|------|-------|------|-------|
| $j$ | 1 | 2 | 3 | 4 | 5 | 6 | 7 | 8 | 9 | 10 |
| $d_\phi^{f(4,j)}$ | 3.498 | 3.266 | 2.546 | - | 3.087 | 2.629 | 2.711 | 3.805 | 3.494 | 3.849 |

**Closest flip point and $\mathcal{B}(x_\phi)$.** The flip point closest to $x_\phi$ is with the class bird, distanced 2.546 from it (measured in L2 norm in the 64-dimensional feature space). This flip point is depicted in Figure 3a, and its distance to $x_\phi$ defines the radius of $\mathcal{B}(x_\phi)$. Any point in feature space that is a member of $\mathcal{B}(x_\phi) \cap \Omega_\phi$ (i.e.,

closer than 2.546 to $x_\phi$) is guaranteed to be classified as Cat by the model. Moreover, Lipschitz constant for the feature space is 6.122, the largest singular value of $W_\phi$, enabling us to study this space with clarity. Intriguingly, we see that 437 training samples and 69 testing samples are actually inside the $\mathcal{B}(x_\phi) \cap \Omega_\phi$ centered at $x_\phi$. We then solve the optimization problem defined by equations 17-18 to find the image in the pixel space that would map to this specific flip point, obtaining the image shown in Figure 3c. This image can be considered the closest adversarial example in $\Phi$, however, in the pixel space, it looks very different from the original image.

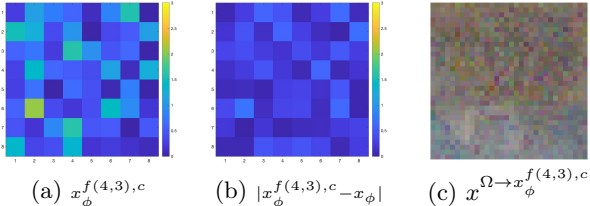

(a) $x_\phi^{f(4,3),c}$    (b) $|x_\phi^{f(4,3),c} - x_\phi|$    (c) $x^{\Omega \rightarrow x_\phi^{f(4,3),c}}$

Figure 3: **(a)** Closest flip point in $\Phi$ for image in Figure 2, **(b)** difference between the closest flip point and $x_\phi$, **(c)** image that would directly map to $x_\phi^{f(4,3),c}$

**Convex hull of training set in feature space.** The fact that some training samples are members of $\mathcal{B}(x_\phi) \cap \Omega_\phi$ implies that the convex hull of the training set overlaps with $\mathcal{B}(x_\phi) \cap \Omega_\phi$. Let us remember that this testing sample, as well as all other testing samples of this dataset, are outside the convex hull of training set, both in pixel space and in feature space. However, geometric arrangements are different in the feature space. In the pixel space, usually, decision boundaries are very close to both training and testing samples. It is known that adversarial examples, i.e., close-by images on the other side of decision boundaries, are so similar to original images that their differences are not easily detectable by human eye. At the same time, in the pixel space, convex hull of training set is rather far from images, and images have to visibly change to reach their $\mathcal{H}^{tr}$. See, for example, Figure 4b for the projection of our first testing sample to the convex hull of the training set in the pixel space, and notice that the image has considerably changed while changes are related to the object of interest as shown in Figure 4c.

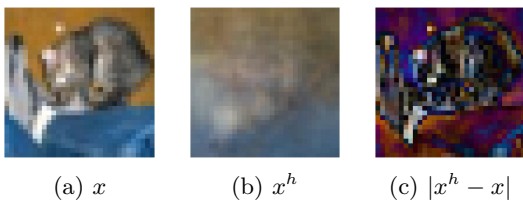

(a) $x$    (b) $x^h$    (c) $|x^h - x|$

Figure 4: First testing sample in CIFAR-10 dataset **(a)** in pixel space, **(b)** its projection to $\mathcal{H}^{tr}$, **(c)** their difference.

In feature space, however, this order is reversed, i.e., convex hull of training set is closer to the sample compared to decision boundaries. Figure 5a shows the projection of our testing sample to $\mathcal{H}_\phi^{tr}$ using equation 24. This point is distanced 0.508 from $x_\phi$, smaller than the 2.546 distance to closest decision boundary in feature space. Notice that the corresponding image in Figure 5c, derived from equations in Section 5, looks more similar to the original image compared to the closest image on the decision boundary shown in Figure 3c and also the projection in the pixel space shown in Figure 4b. Hence, in the feature space, testing sample is more closely related to the convex hull of training set.

**Support in the training set.** Let us now look at training images that participate in the convex combination leading to $x^h$ and $x_\phi^h$. Figure 6a shows four images with largest $\alpha$ coefficients that contribute to the convex hull projection in pixel space, shown in Figure 4b. Coefficients refer to the optimization parameter $\alpha$ in equation 21. Note that only one of these images is from the Cat class while others are from the classes of Automobile, Deer, and Dog.

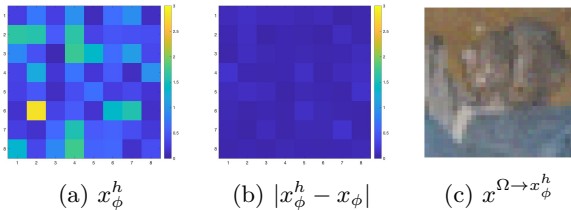

(a) $x_\phi^h$      (b) $|x_\phi^h - x_\phi|$      (c) $x^{\Omega \to x_\phi^h}$

Figure 5: **(a)** Projection of $x_\phi$ to convex hull of training set in feature space, **(b)** difference with $x_\phi$, **(c)** image that would directly map to $x_\phi^h$.

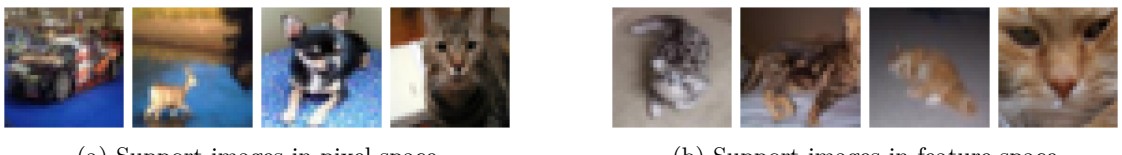

(a) Support images in pixel space      (b) Support images in feature space

Figure 6: Images that form the point on the convex hull of training set, closest to image 2a.

Similarly, Figure 6b shows the training images with largest $\alpha$ coefficients supporting the projection of our image to the convex hull in feature space. These image are all from the Cat class, and the resulting image in the pixel space (Figure 5c) looks more similar to the original image.

**Images on the perimeter of $\mathcal{B}(x_\phi)$.** We seek images in the pixel space that would map to the perimeter of $\mathcal{B}(x_\phi) \cap \Omega_\phi$ in the feature space. This is done by solving the optimization problem defined by equations 15-16 using $r = 2.546$ and with different reference points. To ensure images are on the perimeter, we change the inequality constraint of equation 15 to equality constraint. Finding images on the perimeter of $\mathcal{B}(x_\phi) \cap \Omega_\phi$ can be informative because it shows the extremes of $\mathcal{B}(x_\phi) \cap \Omega_\phi$. Resulting images are shown in Figure 7 next to their reference points.

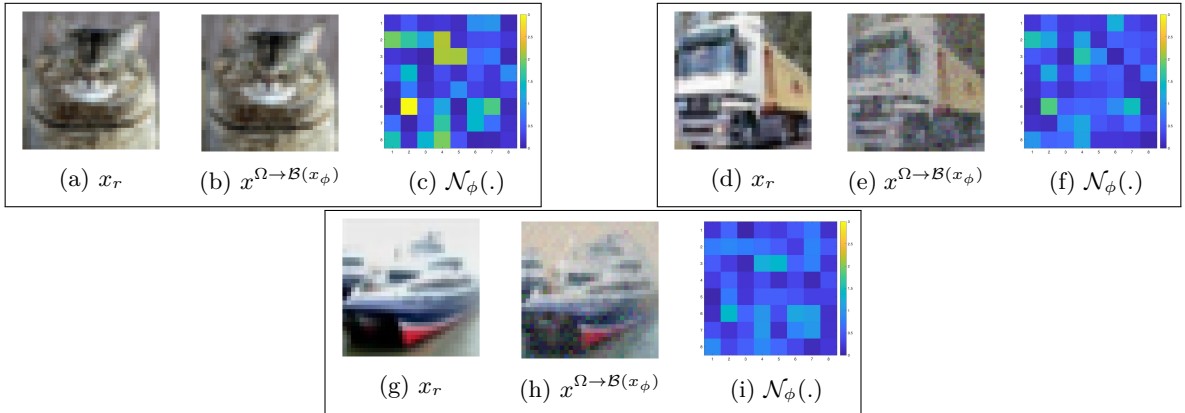

(a) $x_r$    (b) $x^{\Omega \to \mathcal{B}(x_\phi)}$    (c) $\mathcal{N}_\phi(.)$      (d) $x_r$    (e) $x^{\Omega \to \mathcal{B}(x_\phi)}$    (f) $\mathcal{N}_\phi(.)$

(g) $x_r$    (h) $x^{\Omega \to \mathcal{B}(x_\phi)}$    (i) $\mathcal{N}_\phi(.)$

Figure 7: A variety of images in pixel space may map to the perimeter of $\mathcal{B}(x_\phi)$ for a particular image. The second image in each box is on the perimeter of $\mathcal{B}(x_\phi)$ for the first testing sample of CIFAR-10.

**Morphing between images.** We now explore the path between two images inside the $\mathcal{B}(x_\phi) \cap \Omega_\phi$. We pick the image shown in Figure 8a which is the $19821^{th}$ sample from the training set. In $\Phi$, this image is distanced 2.546 from $x_\phi$, so it is close to the perimeter of $\mathcal{B}(x_\phi)$. We gradually move between these image in the feature space and find how the path between them maps back to the pixel space. This is done by solving equations 15-16 while decreasing the value of $r$ from 2.54 to 0. Result is depicted in Figure 8.

Note that moving between these images in the feature space leads to a morphing process between them in the pixel space which is more sophisticated than simple image interpolation (Lakshman et al., 2015).

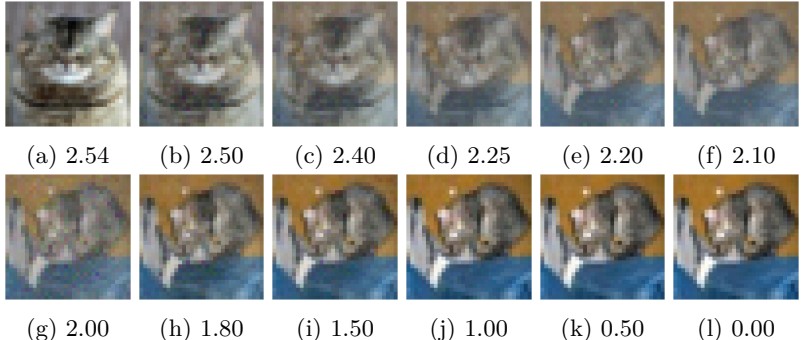

(a) 2.54     (b) 2.50     (c) 2.40     (d) 2.25     (e) 2.20     (f) 2.10

(g) 2.00     (h) 1.80     (i) 1.50     (j) 1.00     (k) 0.50     (l) 0.00

Figure 8: Morphing between two images in the feature space. Distance is measured from the projection of $l$ in feature space. This entire path is inside the $\mathcal{B}(x_\phi)$ and therefore, classified as Cat.

Hence, our formulations can be used for image morphing which has practical applications (Effland et al., 2021). Moreover, this transformation is not linear, i.e., change does not occur at a linear rate along the path between the two images. The image in subfigure $(a)$ is distanced 2.54 from subfigure $(l)$. By the time its distance is 2.2 from $(l)$, it appears more similar to $(l)$ than $(a)$. By the time its distance to $(l)$ is 1.80, it looks almost like $(l)$ despite its relative closeness to $(a)$.

**Mapping paths from the pixel space to the feature space.** In the previous experiment, we moved between two images in the feature space and saw how they morph in the pixel space. Let us now move between those same images in the pixel space and see how the path between them looks like in the feature space. In the pixel space, we follow a direct path along a line connecting these two images, but as Figure 9 shows, the resulting path between them in the feature space is far from a direct line. Our feature space, $\Phi$, is 64-dimensional. To draw this path in 2 dimensions, we use the two-point equidistant projection method as explained in Appendix A.

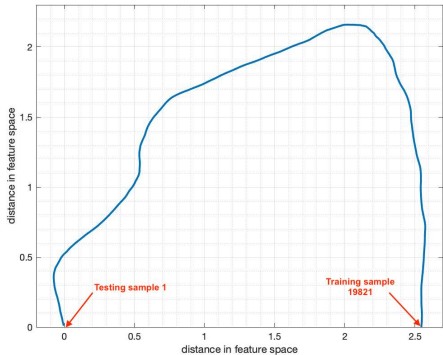

Figure 9: Direct paths in pixel space map to highly curved paths in the feature space. The blue line shows the direct path between images shown in Figures 8a and 8l, mapped to the 64-dimensional feature space, then visualized in 2D.

## 7.2 Larger Trends in the CIFAR-10 dataset

We extend this analysis to the entire dataset to see the larger trends persistent for most images.

**Geometric arrangements in the feature space.** Figures 10a-10b show that for most testing samples, distance to the closest decision boundaries is larger than the distance to convex hull of training set. This difference has broad implications. For example, when we project testing samples to the convex hull of training set in the pixel space, testing accuracy of the model drops from above 90% to 33% on those projected images. However, when we project the testing samples to the convex hull of training set in the feature space, the

accuracy does not change at all, meaning that in the feature space, model has not defined any decision boundaries separating testing samples from their projections to the $\mathcal{H}_\phi^{tr}$.

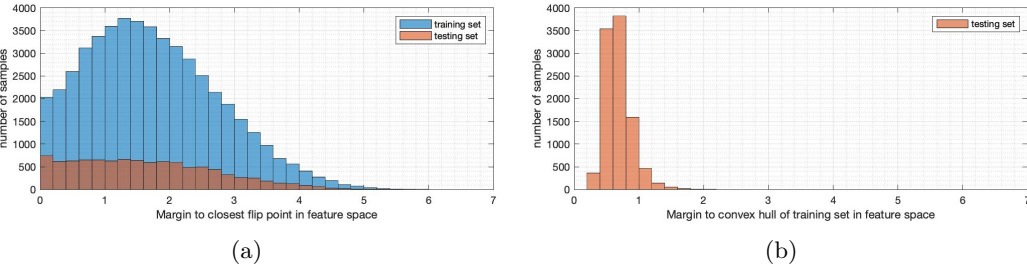

(a)                              (b)

Figure 10: Distribution of distance in the feature space **(a)** to closest flip point, **(b)** to convex hull of training set. For most samples, decision boundaries are much further away than the convex hull of training set. In the pixel space, however, this relationship is reversed.

**Detecting ambiguous images.** In feature space, convex hull of the training set is closer than the decision boundaries for 78.3% of testing samples. Let us see what is different about the remaining 21.7% of images. Testing sample #732, shown in Figure 11a, is distanced 0.3745 from the closest decision boundary in $\Phi$ while its distance to the $\mathcal{H}_\phi^{tr}$ is 2.143. This is clearly an ambiguous image from the model's perspective, because in the feature space, this image is very close to model's decision boundaries, yet very far from the training set.

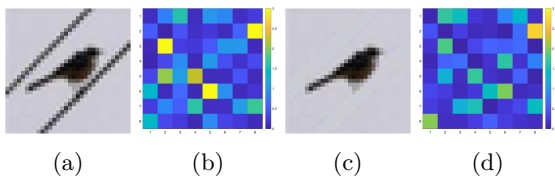

(a)           (b)           (c)           (d)

Figure 11: **(a)** Testing sample #732, **(b)** Mapping of (a) to $\Phi$, **(c)** Modified version of (a) to remove its ambiguity, **(d)** Mapping of (c) to $\Phi$.

From a human's perspective, as opposed to the model's, ambiguity may be perceived differently because a human typically have seen many instances of birds and alike, in different settings/contexts and against various backgrounds. However, the model trained on the CIFAR-10 training set has only seen 5,000 bird images, and the testing image #732 is not similar to any training image regarding the parallel wires below and above the bird. Therefore, this testing image can be considered ambiguous.

Let us now try to remove the ambiguity by eliminating the parallel wires as shown in Figure 11c. Mapping of this modified image to the feature space is drastically different than the mapping of original image. In fact, in $\Phi$, these two images are 5.21 apart which is considerable compared to those distances we previously reported for other images (e.g., in Figures 9 and 10a). The modified image is only distanced 0.605 from the $\mathcal{H}_\phi^{tr}$ while its distance to the closest flip point has drastically increased to 1.225 (the flip point to the Airplane class). From the model's perspective, our modification has removed the ambiguity from the image because now, in the feature space, the image is much closer to the convex hull of training set and it has also moved away from the decision boundaries.

Figure 12 shows the visualized path in $\Phi$ between the testing image #732 and its unambiguous counterpart. As we can see, the path between these images is nonlinear even though moving between them is merely, gradual removal of the wires. But note that non-linearity of the path is more moderate in comparison to the path in Figure 9.

**Formalizing an ambiguity indicator.** This leads us to consider the difference between the distance to closest flip point in $\Phi$ and the distance to $\mathcal{H}_\phi^{tr}$ as a relative indicator for ambiguity

$$d_\phi^{f-h} = d_\phi^{f,min}(x_\phi) - d_\phi^h, \tag{26}$$

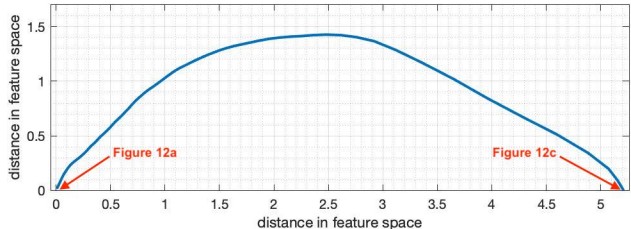

Figure 12: Visualization of direct path between images shown in Figures 11a and 11c.

drawing from the distances previously defined by equations 12 and 25. Figure 13 shows images with extreme values of $d_\phi^{f-h}$.

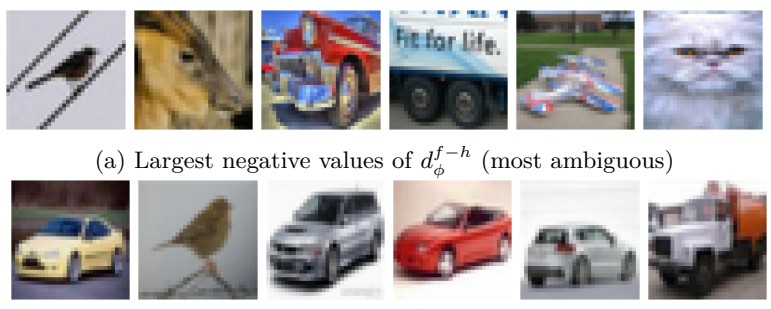

(a) Largest negative values of $d_\phi^{f-h}$ (most ambiguous)

(b) Largest positive values of $d_\phi^{f-h}$ (least ambiguous)

Figure 13: Images with the largest values of $d_\phi^{f-h}$ which we consider to be an ambiguity indicator. Images in (a) are close to model's decision boundaries and far from the convex hull of the training set in the feature space. Images in (b) are far from model's decision boundaries yet very close to the convex hull of training set in $\Phi$.

This notion of ambiguity takes into account closeness to decision boundaries in the feature space learned by the model and contrasts it with the farness from $\mathcal{H}_\phi^{tr}$. When an image falls close to a decision boundary in a feature space, the model may be unsure about the classification, because the image may easily cross the close-by decision boundary and fall into the partition for a different class. Regarding the $\mathcal{H}_\phi^{tr}$, when an image falls close to $\mathcal{H}_\phi^{tr}$, it means that the model has a close point of reference to it in the training set, and therefore, can be more confident in the correctness of classification. By this logic and from the trained model's perspective, all images in Figure 13a can be considered ambiguous while all images in Figure 13b can be considered unambiguous.

We note that most ambiguous images we report are previously reported to be ambiguous for humans from the cognitive science perspective via empirical studies by Peterson et al. (2019); Battleday et al. (2020). In those cognitive science studies, humans were presented with a certain number of training samples of CIFAR-10, and then, were asked to classify testing images of CIFAR-10 in a certain time frame. Ambiguity of images was characterized based on the correctness of human classifications and the time it took for humans to classify them. This can be the subject of further study from the perspective of cognitive science and psychology as well as machine learning. Identifying ambiguous images are also useful in practice.

**Detecting adversarial examples.** Geometric arrangements in the feature space have implications for detecting adversarial images. *Our suggested rule of thumb is that any testing image very close to a decision boundary is likely to be an adversarial input.* For testing samples in this dataset, we see that using a simple threshold, we can detect all adversarial inputs.

Consider, for example, the image in Figure 14a and its adversarial counterpart in Figure 14b classified as Airplane. The original image is distanced 2.494 from the closest decision boundary while its distance to $\mathcal{H}_\phi^{tr}$ is 0.845. On the other hand, its adversarial version is distanced 0.0001 from the closest decision boundary

in $\Phi$ while its distance to $\mathcal{H}_\phi^{tr}$ is 0.640. Extreme closeness of this sample to the decision boundary in $\Phi$ is a clear indication of its adversary nature.

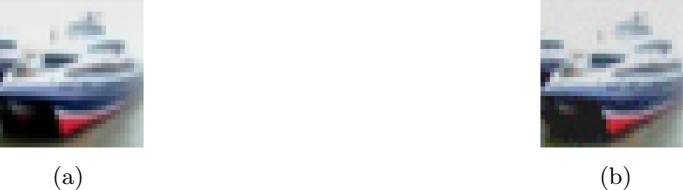

(a)                                             (b)

Figure 14: **(a)** Testing sample #2, **(b)** Adversarial version of it classified as Airplane.

In the pixel space, genuine images are often very close to decision boundaries, so closeness to decision boundaries is not a good measure to distinguish adversarial inputs from genuine ones. In the feature space, however, decision boundaries are relatively far from the images, especially in comparison to the $\mathcal{H}_\phi^{tr}$. In other words, an image very close to the decision boundaries of feature space is unusual, and this closeness can be used as an indicator.

For 100% of testing samples, their adversarial version is closer to the decision boundaries of the feature space compared to the $\mathcal{H}_\phi^{tr}$. Their margin to decision boundaries is also closer than the margin to decision boundaries for all training/testing samples. In other words, adversarial methods move the testing samples, recognizably, very close to the decision boundaries of the feature space, by any of these measures of comparison.

Moreover, we see that standard adversarial methods such as DeepFool move the samples towards the $\mathcal{H}_\phi^{tr}$. See Appendix B for further discussion.

**Out-of-distribution detection.** Using the ambiguity measure in equation 26, we evaluate the ambiguity of MNIST images for our model trained on the CIFAR-10 dataset. Figure 15 shows the ambiguity measure for the testing images of MNIST vs the ambiguity measure for training and testing sets of CIFAR-10 dataset.

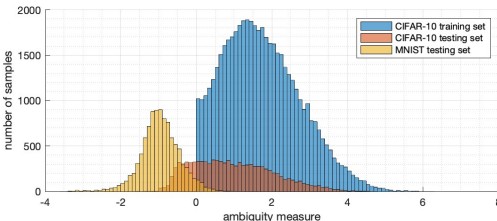

Figure 15: Ambiguity of MNIST and CIFAR-10 images for our model. Using the threshold of 0 for $d_\phi^{f-h}$ our model can separate 96% of MNIST images as ambiguous and abstain from classifying them.

As we can see, our ambiguity measure considers almost all images of MNIST dataset as highly ambiguous. One can use the threshold of 0 for $d_\phi^{f-h}$ to identify images that are ambiguous. Using this threshold, more than 96% of MNIST images are ambiguous while all training samples of CIFAR-10 are unambiguous. About 80% of testing samples of CIFAR-10 would also be considered unambiguous. In a practical setting, the ambiguity measure can be used for a model to abstain from classification and/or flag such inputs for review by humans.

**Union of learned regions.** Earlier, we mentioned that a classification model is a function defined by its decision boundaries. A model learns from the contents of training images and their labels. Via this process, model partitions the domain (in pixel space and in feature space) by defining certain decision boundaries. We defined the ball around each image that borders with the closest decision boundary. Such ball can be viewed as a region known to the model and guaranteed to have a certain classification. We reported earlier that for the first testing sample of dataset, the radius of that ball was quite large in the feature space such that it contained hundreds of training and testing samples, having a considerable overlap with the convex hull of training set. This trend holds for many other images in the dataset. In fact, for 49.9% of training

samples, their corresponding $\mathcal{B}(x_\phi)$ contains at least another training or testing sample. Similarly, for 47% of testing samples, their $\mathcal{B}(x_\phi)$ contains other training/testing samples. On average, each $\mathcal{B}(x_\phi)$ contains about 297 other training and testing samples. The largest number of samples contained in a $\mathcal{B}(x_\phi)$ is 4,791.

Therefore, the learned regions, defined by $\mathcal{B}(x_\phi)$ around each image, have significant overlaps, and we can study the *union of learned regions* defined by

$$\bigcup_{i=1}^{n} \mathcal{B}(x_i),$$

for all the $n$ samples in a training set.

Overall, more than 68% of testing samples are contained in the union of learned regions for the training set. These samples could be considered most familiar samples for the model as they fall into familiar regions in the feature space relating closely to training samples. This concept may also be useful for detecting out-of-distribution images, and images with low-confidence in their classification.

**How images are supported by the convex hull of training set.** Using equation 24, we project each testing image to the convex hull of training set. We perform this both in the pixel space and in the feature space. Projection of each image to the convex hull is a point defined as a convex combination of certain support images in the training set. We see that in the feature space, 78% of support images have the same label as the testing image that they are supporting while this percentage is only 27% in the pixel space. This shows that in the feature space, a testing image of a given class, let us say Automobile, is supported mostly by training images of Automobile class, whereas in the pixel space, a testing image from the Automobile class may be supported by training images of many other classes. This is another evidence that geometric arrangement of images in the feature space is more sensible and meaningful from the classification perspective.

## 8 Conclusions

In this work, we presented a set of formulations that can answer questions about the inner workings of feature space learned by trained neural networks. Our formulations incorporate any trained model as a function, and find images in the pixel space that map to particular points and regions of interest in the feature space. This enabled us to provide many novel insights about image classification functions, the features that they learn from images, and their adversarial vulnerabilities. Although our formulations about the pixel space are generally hard to solve, we were able to solve them with a homotopy algorithm. The feature space, on the other hand, is Lipschitz continuous with a known constant which enable us to study it with clarity. We identify certain regions around each image guaranteed to have the same classification as the image. We then investigated these regions with respect to training samples and decision boundaries of the model. Notably, we observed that geometric arrangements of decision boundaries are considerably different in the feature space in relation to training and testing samples, providing a way to identify ambiguous and adversarial images. These geometric arrangements are very different than the ones reported in the literature for the pixel space. Moreover, a new direction of research would be to study adversarial examples that remain far from the decision boundaries and at the same time, maintain a distance from the convex hull of training set in feature space. Finally, these insights may inform us about the functional task of models and the extent in which they extrapolate to classify unseen images.

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
