# OpenReview forum: "Decision Boundaries and Convex Hulls in the Feature Space that Deep Learning Functions Learn from Images"
_TMLR — Rejected by TMLR_

### Review · Reviewer_4f3L · 2022-05-25

**Summary Of Contributions:**

The paper suggests several techniques for exploring the behavior of decision boundaries in feature space and pixel (input) space of a trained neural network. The feature space considered is the input to the last layer of a neural network. Then, quantities like the margin of a point (distance to decision boundary) can be solved accurately as it can be formulated as a convex problem. In addition, it also suggested exploring areas in input space that get mapped to interesting points in feature space. This technique does not have any guarantees as it is formulated as a global/non-convex optimization problem.

**Broader Impact Concerns:**

A broader Impact Statement is not provided. I do not have any concerns regarding this issuse.

**Requested Changes:**

All the weaknesses stated above should be considered in the next revision of the paper.

**Strengths And Weaknesses:**

Strengths
----------

Some interesting findings regarding the differences between the pixel space to the learned feature space are presented. In particular, Figure 6 shows that a convex hall of training samples in feature space is related to test examples in contrast to the input space.

Weaknesses
--------------

quality of presentation. The abstract and intro require a revision to be more concise. In particular, a long list of previous approaches to learned feature analysis is presented, but I could not understand what these methods are missing and what the current paper comes to fix/add. Another example is the contribution list at the end of the intro. I believe this list should better represent the contributions of the paper: e.g. the discussion on inverse problems and the usage of some algorithm does not seem to lie at the core of the contributions. More importantly, it is not stated why the findings in the paper are important and what are their implications. E.g., it is stated that decision boundaries and convex hulls in feature space differ from the pixel space. What is meaningful about this? why is it not trivial?

unclear insights. It is claimed that Figure 2 and Figure 3 provide some insights. However, I could not find in the text what insight they provide. It is not clear why it is interesting to show the image in feature space Figure 2 (b). Also why it is important to show its margin to other classes? More importantly, what do we learn from figure 3? even the paper itself is reluctant about finding images in input space that satisfy some condition (inverse problem) as there could be many solutions to the inverse problem. Whey do we learn something from this particular solution exhibited in figure 3 (c)?

The idea of using feature space for morphing applications could be interesting but is presented in a crude fashion. How does it work compared to an autoencoder (That does not require labels)? The same comment is also relevant to identifying adversarial examples using margin in feature space. How does it work as a defense?

The motivation for analyzing the properties of a model with respect to the convex hull of samples is not stated in the paper. What do we learn about a model if samples can be represented as a convex combination in some space? this should be discussed in the paper.

Figure 1 is a bit misleading. If I understand correctly, the decision boundaries in feature space are linear (straight lines). However, figure 1 is showing non-linear decision boundaries corresponding to the input space.

---

### Review · Reviewer_oeJ4 · 2022-06-02

**Summary Of Contributions:**

This paper reports findings from an empirical study of the geometry of the feature space of one convolutional neural network architecture trained on CIFAR-10. The main objects of study are the distances of the train/test points to the decision boundaries in feature space, as well as the convex hull of the training data. Through standard optimization techniques akin to the ones used in prior feature visualization studies, the authors propose to map important *loci*  in this feature space (such as the minimal invariant ball around a feature vector) to other points in the pixel space close to certain references.

The main reported findings of this work (validated only in the context of their experiments) are:
1. Most typical samples lie closer to the convex hull of data in the feature space than to the decision boundaries. This is the opposite effect as the one observed in the input space.
2. Adversarial examples (computed in the input space through standard algorithms) also lie very close to the decision boundaries in the feature space.
3. In an MNIST OOD detection task, the difference in distance to the feature convex hull and the decision boundary can be used to accurately detect MNIST examples (i.e., OOD) using the feature space of the CNN under study.
4. Ambiguous examples (i.e., difficullt to classify by the network) lie closer to the decision boundary than the convex hull.

To the best of my knowledge, this very specific observations have not been reported before.

**Broader Impact Concerns:**

I do not believe there exist major impact concerns directly linked to this work.

**Requested Changes:**

To recommend acceptance of this work I would require to see a major overhaul of the experimental sections. In particular, I would need to see:
1. All findings replicated for other standard architectures, datasets, and hyperparameter choices (e.g., choice of $x_r$).
2. I would also need a thorough benchmarking of the proposed OOD detection techniques in line with [current practices](https://arxiv.org/pdf/2006.10108.pdf).
3. In terms of adversarial example detection, it is important that the results are properly described and thoroughly reported. Giving aggregate numbers in-text without a proper description does not provide any scientific support. I would expect a more quantitative and in-depth analysis that can go beyond subjective comments like "*Our suggested rule of thumb is that any testing image very close to a decision boundary is likely to be an adversarial input*". Otherwise, I would refrain from including these results.
4. A clear discussion of the actual contributions of this work in comparison to the vast current feature visualization literature. Why do we need the methods proposed in this work if there exist other approaches to perform the same tasks?

On the other hand, I am of the opinion that some restructuring of this work, so that the main messages are more streamlined, even at the expense of removing content, would make this submission much stronger.

**Strengths And Weaknesses:**

## Strengths
1. **Relevant topic**: The problem of understanding how neural networks construct their feature space, and describing its geometry is an interesting topic for scientific work.
2. **Technical correctness**: The technical tools followed by this work to estimate distances and decision regions in that space seem correct.

## Weaknesses
1. **Methodology**: In my opinion the main weakness of this work is that it does not follow a rigorous scientific methodology to validate the generality of its findings. It is quite likely that the reported results are factually true, but without proper investigation of alternative valid hypothesis, thorough benchmarking, and relying very heavily on cherry-picked results, these findings become nothing more than just anecdotal evidence. Let me give examples of these behaviours that can be found throughout the paper:

- **Cherry-picking**: Multiple times throughout the work, the authors make claims about specific measurements taken on a single cherry-picked example and directly extrapolate to the whole population. For example, in section 7.1. the authors say that the projection of images to the convex hull in feature space *visually* changes an image less than its projection to the decision boundary. This is first, a subjective statement, and second it is only shown for one example, for one architecture, trained on one dataset, and for a single seed.

- **Arbitrariness**: Many technical decisions of this work seem rather arbitrary and are not properly supported in the text. For example, why this specific choice of architecture? Or a 64-dimensional feature space? How do the authors choose the specific reference points $x_r$ in their optimizations.

- **Limited depth of experiments**: This work in my opinion would have been much stronger if it had focused on depth, rather than breadth of experiments. With this I am referring to the fact that the authors are trying to cover a lot of empirical ground and try to provide insights on many fronts such as OOD detection, adversarial robustness, model inspection... However, reading this work it feels as though none of these topics has been thoroughly studied. For example, the claims that the difference between the distance to the convex hull and the distance to the decision boundary can be used to detect OOD inputs could be very interesting. Yet again, because this work does not follow a thorough evaluation protocol nor tests this claim with respect to the standard benchmarks in that field, the reported findings remain as mere anecdotal evidence.

- **Lack of ablations**: Some claims of this work are not properly ablated. For example, the authors claim that the difference between the distance to the convex hull and the distance to the decision boundary is an important metric that can be used for important problems such as detecting OOD samples, characterizing adversarial examples, or identifying difficult samples. However, no ablation is performed to make sure that it is really the difference in distance what matters. Why not only the distance to the decision boundary? Or the distance to the convex hull alone? These other distances should definitely be compared with the proposed metric before arriving to any conclusion.

2. **Missing references to feature visualization literature (novelty issues)**: The authors claim as one of their novel contributions that they provide computational methods to map feature vectors to images in the input space. However, this precise problem has been the study for many years of the [feature visualization community](https://distill.pub/2017/feature-visualization/). Yet, the authors do not seem to acknowledge any work on that field, and I strongly believe they should highlight what are the exact additional contributions that they bring to the table.

3. **Paper structure**: In my opinion the paper structure could be heavily improved. Right now there is a lack of a coherent story to guide the reader throughout the text, and the current manuscript reads more like a collection of independent observations which exacerbates the feeling of arbitrariness.

---

### Review · Reviewer_TT7G · 2022-06-06

**Summary Of Contributions:**

This paper studies the decision boundary and related geometric properties in both feature space (the second to the last layer) and the pixel space. It provide both qualitative investigations and quantitative measurements. The observations were related to other important topics in deep learning such as adversarial vulnerability.

**Requested Changes:**

1. Statistical significance of the result by considering repeated random runs (see above).
2. Organization of the paper (see above).
3. It would be useful to have a related work section.

**Strengths And Weaknesses:**

**Strength**: This paper studies the geometric properties of learned decision boundaries and relate to a number of potential applications such as outlier detection and adversarial examples.

**Weakness**:
The methodology used in this paper looks correct, though I would like to see the experiments repeated in different runs. Since there are a lot of randomness in neural network training, the final network weights are generally very different on different runs. Are the decision boundaries also different? Or are they relatively stable? Do the general observations consistently hold across repeated runs?

Another aspect that I think this paper could improve on is the presentation. Currently it define all the terms and introduce a lot of notations that are similar to each other. Then in the second part, it presents qualitative results, and in the last part quantitative results. It is quite hard to follow the narration and not being confused about what the terms are referring to. One way to improve this is to only define the terms when the relevant results are presented. In other words, divide the paper according to different aspects, and then put definitions, quantitative and qualitative results related to each aspect together. Also, many notions are quite standard, and can be described with much lighter notations or even completely in words (e.g. projection onto convex hull).

---

### Decision · Action_Editors · 2022-07-30

**Recommendation:** Reject

**Comment:**

Reviewers were unanimous in their tendency to reject the manuscript, highlighting issues such: lacking scientific methodology in support of raised claims, unclear delineation of the work's contributions (with references to important existing works missing) and unclear presentation.  Given that the authors chose not to respond (despite being given an extension of several weeks), I have no choice but to recommend rejection.

---

> ### Author Response · Authors · 2022-08-15
> **Thank you**
>
> Dear Editors and Reviewers,
>
> We are writing to thank you for reviewing our paper. We appreciate your thoughtful comments and suggestions. Due to a serious of unforeseen events, unfortunately, we were not able to engage with the reviewers and address their concerns during the discussion period - we are sorry about that. We understand the concerns of reviewers and the following decision.
>
> We are in the process of expanding the experiments to more models. New experiments are supportive of our original results. We plan to improve the paper and submit a revised version at a later time.
>
> Sincerely,
> Authors